



# The polarimetric characteristics of dust with irregular shapes: Evaluation of the spheroid model

Jie Luo[1,2], Zhengqiang Li[1,2], Cheng Fan[1], Hua Xu[1,2], Ying Zhang[1,2], Weizhen Hou[1,2], Lili Qie[1], Haoran Gu[1,2,3], Mengyao Zhu[1,2,4], and Yinna Li[1,2]

[1]State Environment Protection Key Laboratory of Satellite Remote Sensing, Aerospace Information Research Institute, Chinese Academy of Sciences, Beijing 100101, China
[2]University of Chinese Academy of Sciences, Beijing 100049, China
[3]College of Geography and Tourism, Anhui Normal University, Wuhu 241003, China
[4]College of Geoexploration Science and Technology, Jilin University, Changchun 130026, China

**Correspondence:** Zhengqiang Li (lizq@radi.ac.cn)

**Abstract.**

In the atmosphere, the dust shapes are various and a single model is difficult to represent the complex shapes of dust. We proposed a tunable model to represent dust with various shapes. Two tunable parameters were used to represent the effects of the erosion degree and binding forces from the mass center, respectively. Thus, the model can represent various dust shapes

by adjusting the tunable parameters. To evaluate the applicability of the spheroid model in calculating the optical properties, the aspect ratios of spheroids were retrieved by best fitting the phase function of dust with irregular shapes. Our findings show that the dust shapes have a substantial impact on the scattering matrix, and sometimes the sign of elements of the scattering matrix can be modified by changing the tunable parameters. The applicability of the spheroid model is significantly affected by the erosion degree and binding forces, and substantial deviations could be observed when the dust size is relatively large.

Besides, the sign of $F_{12}/F_{11}$ and $F_{34}/F_{11}$ can be modified from negative to oppositive at some scattering angles if substituting the irregular dust with best-fitted spheroids. As the binding force is small, the deviation of extinction/scattering cross-section generally increases with the erosion degree, and the relative difference can reach approximately 30% when the erosion degree is large, while the deviation is mitigated with the binding force increasing. Besides, with the binding force increasing, the retrieved aspect ratio is more close to 1:1. The deviations of the spheroid model on estimating the polarized light were also

investigated using the successive-order-of-scattering (SOS) vector radiative transfer (VRT) model. With a diameter ($d_p$) of 0.2 $\mu$m, the relative difference of normalized radiance does not exceed 3%, and the absolute values of the deviation of the polarized bidirectional reflectance factor (PBRF) and the ratio of radiance to polarized intensity (DoLP) are below 0.005 and 0.02, respectively. However, with the particle size increasing, the difference becomes much more substantial. The relative difference of the normalized radiance can exceed 10%, and the deviation of the PBRF and DoLP can vary in the range of

-0.015 - 0.025 and the range of -0.05 to 0.15. Thus, the use of the spheroid model in the component retrievals based on the polarized light should consider the effects of more complex dust shapes.



## 1    Introduction

Dust particles, as a main atmospheric aerosol in the earth system, play an important role in climate forcing (IPCC, 2014; Textor et al., 2006). Dust can direct absorb and scatter solar radiation, thereby modifying the radiation balance (Huang et al., 2009; Ge et al., 2010; Li et al., 2016). Dust can also modify the cloud properties by serving as the cloud condensation nucleus (CNN), so play an indirect effect on the climate (Li et al., 2010; Seigel et al., 2013). Besides, as an important aerosol in the atmosphere, dust is a main source of PM2.5 and PM10 (Kuo and Shen, 2010), and it can significantly affect the air quality. Thus, the monitoring of the dust in the atmosphere can improve the understanding of the drivers of climate change and air quality.

Remote sensing, as an effective tool for monitoring earth, has been applied to retrieve the aerosol properties (Kokhanovsky et al., 2007; Dubovik et al., 2006; Zhang et al., 2020; Si et al., 2021). Ground-based remote sensing and satellite remote sensing are the main techniques to retrieve aerosols. Ground-based remote sensing, such as the AERONET (AErosol RObotic NETwork) project (Holben et al., 1998), mainly inverting the aerosol properties based on the optical measurements from the sun-Sky-Lunar spectral photometer (Dubovik et al., 2002; Li et al., 2018b; Sinyuk et al., 2020), and it can provide relatively accurate estimations. However, ground-based remote sensing is difficult to cover the global range. Satellite remote sensing allows us to see a much larger area than ground-based remote sensing, so it can provide regional/global measurements. However, satellite remote sensing may provide inaccurate estimations due to the incomplete understanding of the optical properties of aerosols. The traditional satellite aerosol retrieval algorithms mainly derive the whole floor of aerosols based on the radiation fluxes, while it is difficult to estimate the contribution of different components due to the perturbs of the surface.

The polarization is more sensitive to the atmosphere compared to the surface, and suffers fewer perturbs from the surface (Dubovik et al., 2019; Li et al., 2018a). Thus, previous studies have applied polarimetric remote sensing to monitor atmospheric aerosols (Dubovik et al., 2019; Hasekamp et al., 2011; Dubovik et al., 2011; Xu et al., 2016). Both traditional remote sensing and polarimetric remote sensing require a forward model for radiative transfer calculations, and we need to provide the optical properties of aerosols as input for the radiative transfer model. The extinction coefficient, single-scattering albedo, and phase matrix are the most important aerosol optical parameters for radiative transfer calculations (Mishchenko et al., 2002; Liu and Mishchenko, 2005; Heidinger et al., 2006). In remote sensing based on the radiative fluxes, for efficient calculations, most radiative transfer calculations ignore the polarized components, so the phase matrix is simplified as phase function (Lenoble et al., 2007). However, the polarimetric remote sensing is commonly based on the vector radiative transfer calculations, which requires the complete phase matrix of aerosols (Spurr, 2006; Cai et al., 2006). In most remote sensing algorithms, aerosol shapes were commonly assumed to be spherical, and the optical properties can be calculated using the Mie theory. However, the transmission electron microscopy (TEM) and scanning electron microscope (SEM) images have shown that most aerosols exhibit distinct non-spherical shapes (Lindqvist et al., 2014; Woodward et al., 2015; Chou et al., 2008; Luo et al., 2021a, 2018a, b, 2021b). Dust particles are the typical non-spherical particles (Lin et al., 2018; Dubovik et al., 2006, 2011).

The spheroid model, as one of the most simplified non-spherical models, has been used to model the optical properties of dust (Dubovik et al., 2006, 2011). Compared to the sphere model, the aspect ratio needs to be determined. The retrieval algorithm based on the measured phase matrix was developed (Dubovik et al., 2006, 2011). However, in the atmosphere, the



dust exhibits much more complex shapes than spheroids (Lindqvist et al., 2014; Woodward et al., 2015; Chou et al., 2008; Lin et al., 2018), and the spheroid model is difficult to simultaneously best fit all the elements of the phase matrix (Lindqvist et al., 2014). The first element of the phase matrix (phase function) is most important for radiation calculations. In traditional remote sensing, the polarization was commonly neglected, so the phase function was often used to retrieve the aspect ratio

of spheroids. For example, Mishchenko et al. (1997) have used the phase function measurements to retrieve the aspect ratio of dust spheroids; Kocifaj et al. (2008) have retrieved the aspect ratio based on the phase function and extinction coefficient simultaneously. Li et al. (2019) have shown that the polarimetric characteristics calculated assuming the microscope-measured aspect ratio is distinct from that using the inversion-based aspect ratios. Nevertheless, in the aerosol component retrievals, we pay much less attention to the retrieved aspect ratio. We mainly focus on the retrieved size distribution and refractive index of

aerosols. To accurately retrieve the size distribution and refractive index of dust particles, we should calculate accurately the optical properties of dust particles under the real size distribution and refractive index, but the aspect ratio may be not identical to the microscope-measured aspect ratio. It is still unclear that whether the spheroids with the retrieved aspect ratio based on the phase function can represent the polarization of dust with more complex shapes.

Some modeling works have been conducted to investigate the optical properties of more irregular dust, and they have shown

that the optical properties of dust are significantly affected by their shapes (Yang et al., 2007; Lindqvist et al., 2014; Bi et al., 2010; Liu et al., 2013; Escobar-Cerezo et al., 2017; Zubko, 2013; Kanngießer and Kahnert, 2021; Kemppinen et al., 2015; Kahnert, 2015). However, in the previous studies, the computations were only conducted by assuming some specific shapes. In the atmosphere, the dust shapes are various, and a single model is difficult to represent the complex shapes of dust, so we need to develop dust models which can represent various shapes. Besides, even though many studies have compared the scattering

properties of spheroids and dust with more complex shapes (Lin et al., 2018; Lindqvist et al., 2014), in their studies, the whole aspect ratio was commonly assumed to be the same as the spheroids, and it is still unclear whether the spheroid model with other aspect ratios can reproduce optical properties of dust with more complex shapes. Besides, the polarimetric sensitivities to the dust shapes were few investigated.

In this work, we attempt to answer the following questions:

– How do we use models to represent dust with various shapes?

– How do the dust shapes affect the polarimetric characteristics?

– Could the spheroid model reproduce the polarization of dust with irregular shapes?

To answer the above questions, we proposed an irregular model to represent the dust with various morphologies, and the scattering properties were calculated using discrete dipole approximation (DDA) methods. Then, we retrieved the aspect ratio

that best fits the phase function of dust with complex morphologies using the spheroid model, and the phase matrices of dust with complex morphologies and spheroids were compared. Besides, the radiance and polarization were calculated using a vector radiative transfer (VRT) code based on plane-parallel successive-order-of-scattering (SOS), and the capabilities of spheroids for representing the radiance and polarization of irregular dust were evaluated.


## 2  Method

### 2.1  Modeling of dust with irregular shapes

To model the dust with irregular shapes, we proposed a model based on the physical process. To evaluate the applicability of spheroids as dust turns more irregular, we assumed that the ideal dust particles are spheroids, but they could become more irregular in the atmosphere. We assumed that the evolution of dust shapes is mainly affected by two factors. On the one hand, the dust could be eroded under the effects of external forces, such as wind, water, etc. Under the erosion of the external forces, the mass of the dust would be lost. On the other hand, the binding force could constrain the loss of dust mass. We have generated various dust shapes based on the above mechanisms.

Firstly, we generated spheroids with different aspect ratios, and they were discretized into numerous dipoles. Then, the dipoles were gradually lost under the erosion of external force. From physical points, the surface of the dust is more easily eroded, and the mass close to the surface may be easier to be lost. To reflect the erosion of external force, we first identified the edge dipoles (close to the surface) occupied by the dust, and then decide if a selected dipole were lost based on a parameter:

$$\mathrm{J} = \frac{1}{N_s} \sum_{i=1}^{N_e} \frac{Rn}{(l_i + 10^{-9})^2} - \frac{R}{l_0{}^2} \qquad (1)$$

where the first item on the right side of the formula reflects the effects of external force. $l_i$ represents the distance from the selected dipole to the $i$th edge dipoles, and larger $1/l_i^2$ denotes a larger external force. Here $10^{-9}$ was used to prevent a zero denominator. $R_n$ is a random value from 0 to 1, which represents the randomness of external force, and it can simulate the roughness of the surface. $N_e$ represents the number of edge dipoles. In this work, as there are too many edge dipoles, to speed up the calculation, we randomly selected 1/5 of the edge dipoles. The second part reflects the binding force from the center of mass. $l_0$ represents the distance from the center of the selected dipole to the mass center of the dust, and larger $1/l_0^2$ denotes a larger binding force. $R$ is a tunable parameter to represent the magnitude of the binding force. Larger $R$ may lead to more spherical dust shapes. The dipoles with larger $J$ indicate that the dipoles were affected by larger external force or small binding force. Thus, the dipoles with larger $J$ are easier to be lost.

We first sort the $J$ value of the dipoles occupied by the dust, and the dipoles with larger $J$ are easier to be lost. With the erosion, the mass of the dust is gradually lost. We define a parameter to represent the ratio of the lost volume to the original dust volume:

$$f = \frac{V_{Lost}}{V_0} \qquad (2)$$

where $V_0$ represents the volume of the original spheroids, and $V_0$ denotes the volume lost in the erosion process. As shown in Figure 2, with a larger R, the dust shapes are easier becomes spherical due to larger binding force. In our algorithm, the R and $f$ are two tunable parameters to reflect the effects of the binding force and erosion degree, respectively, and various dust shapes could be generated by adjusting these two parameters.



## 2.2 Calculation of the single scattering of dust

The normalized scattering matrix, extinction cross-section ($C_{ext}$), and scattering cross-section ($C_{sca}$) are the key parameters to reflect the single scattering properties of aerosols (Mishchenko et al., 2002; Liu and Mishchenko, 2005). To reflect the Stokes vector of polarization, the normalized Stokes scattering matrix has six independent elements (Paton, 1958; Mishchenko et al., 2002):

$$
F(\theta) = \begin{bmatrix} F_{11}(\theta) & F_{12}(\theta) & 0 & 0 \\ F_{12}(\theta) & F_{22}(\theta) & 0 & 0 \\ 0 & 0 & F_{33}(\theta) & F_{34}(\theta) \\ 0 & 0 & -F_{34}(\theta) & F_{44}(\theta) \end{bmatrix} \tag{3}
$$

The first element of the scattering matrix $F_{11}(\theta)$ is the phase function and satisfies:

$$
\frac{1}{2} \int_{o}^{\pi} F_{11}(\theta) sin(\theta) d\theta = 1 \tag{4}
$$

In this work, we mainly focus on the polarization of the dust particles, so the vector radiative transfer equations need to be considered, and the complete stokes scattering matrix was inputted into the radiative transfer equations.

The T-matrix method has great advantages in calculating the optical properties of symmetrical particles (Mishchenko et al., 1996; Kahnert, 2013). In this work, the T-matrix code developed by Mishchenko and Travis (1998) was used to calculate the single scattering properties of spheroids. However, for dust with more complex shapes, such as the dust models proposed in Section 2.1, the T-matrix code of Mishchenko and Travis (1998) is inapplicable. The discrete dipole approximation can calculate the optical properties of particles with arbitrary shapes, and it was used to calculate single scattering properties of the irregular dust particles. In this work, a widely used DDA code, DDSCAT version 7.3, was applied (Draine and Flatau, 2008, 1994), and the first element of the scattering matrix was normalized to satisfy Equation 4. We assumed that the dust particles are randomly oriented, so we average the DDA calculations over $12 \times 7 \times 12 = 1008$ directions, which can achieve relatively accurate results (Dong et al., 2015; Kahnert, 2017; Luo et al., 2019, 2021b). For accurate calculations, the dipole spacing ($d$) satisfies $|m|kd < 0.5$, where $m$ is the refractive index of dust. In this work, the polarimetric characteristics of dust with irregular shapes were investigated at a wavelength of 670 nm, which is a typical polarimetric band in polarimetric instruments/satellites, such as POLDER-1/ADEOS I, POLDER-3/PARASOL, MAI/TG-2, CAPI/TanSat, DPC/GF-5. We assumed the refractive index of dust is 1.52 + 0.005i based on previous studies (van Beelen et al., 2014; Dey et al., 2006). As shown in Figure 3, the difference of the scattering matrix of spherical particles calculated using the DDSCAT is below 1%, and the accuracy of the DDSCAT is acceptable.





### 2.3 Retrieval of the aspect ratio of irregular dust

In this work, we attempt to find spheroids that best fit the phase function of irregular dust particles. Firstly, the scattering matrix of dust with irregular shapes was calculated using the DDSCAT, then the spheroid model was used to retrieve the aspect ratio by minimizing the following function:

$$D = |F_{11\_Irregular} - F_{11\_spheroid}|^2 \tag{5}$$

where $F_{11\_Irregular}$ and $F_{11\_spheroid}$ are the phase function of dust with complex shapes and spheroids, respectively. By minimizing $D$, we can find the aspect ratios that best fit the phase function of dust with irregular shapes.

### 2.4 Radiative transfer calculation

**Table 1.** Input parameters for radiative transfer calculation.

| Paramter | Value |
|---|---|
| Wavelength | 0.67 $\mu$m |
| Aerosol Optical Delpth | 0.2785 (Li et al., 2019, 2018b) |
| Molecular Optical Depth | 0.015 (Lin et al., 2018) |
| Solar Zenith Angle | 45° |
| Surface Albedo | 0.15 |

A successive-order-of-scattering (SOS) vector radiative transfer (VRT) code was employed to calculate the radiance and polarization (Lenoble et al., 2007). The $C_{ext}$, $C_{sca}$ and scattering matrix of dust with different shapes were inputted to the sos model. The polarized light can be characterized by the Stokes vector $[I, Q, U, V]$ . The normalized radiance (I) was widely used to represent the characteristics of bidirectional reflectance. Given the cosine value of the solar zenith angle ($\mu_0$) and the extraterrestrial solar irradiance ($F_0$), the normalized I can be calculated using (Lenoble et al., 2007; Zhai et al., 2013):

$$Normalized\_I = \frac{\pi I}{\mu_0 F_0} \tag{6}$$

Similar to the radiance, the polarized bidirectional reflectance factor (PBRF) was also investigated. PBRF is defined as the normalized polarized intensity, can be expressed as (Zhai et al., 2013; Zhang et al., 2021):

$$PBRF = \frac{\pi \sqrt{Q^2 + U^2}}{\mu_0 F_0} \tag{7}$$

note here we don't consider the circular polarization (V) as the V is commonly small enough.



Another important parameter (DoLP), which characterizes the ratio of radiance to polarized intensity, was also used in polarimetric remote sensing. DoLP is defined as (Li et al., 2019; Zhang et al., 2021):

$$\mathrm{DoLP} = \frac{\pi\sqrt{\mathrm{Q}^2 + \mathrm{U}^2}}{\mathrm{I}} \tag{8}$$

## 3  Results

### 3.1  Single scattering properties of dust with irregular shapes

#### 3.1.1  Effects of dust shapes

The scattering matrices of dust with different shapes are shown in Figures 4 - 6. When the particle size is small ($\mathrm{d_p} = 0.2\mu\mathrm{m}$), the changes of the $\mathrm{F_{11}}$, $\mathrm{F_{33}}$, $\mathrm{F_{44}}$, $\mathrm{F_{12}}$, and $\mathrm{F_{34}}$ are relatively small with the particle shape varying. However, with the particle size increasing, the effects of particle shapes on the scattering matrix become more significant. Fixing the original aspect ratio to 2:1 and the parameter R to 0, with the erosion of the external force (increasing $f$), the phase function exhibits larger forward scattering and smaller backward scattering. With the $f$ varying, obvious variations are observed from $150° - 180°$ scattering

angles. The $\mathrm{F_{22}/F_{11}}$ is also significantly affected by varying $f$. Fixing R to 0, larger $f$ generally leads to smaller $\mathrm{F_{22}/F_{11}}$. This means that the erosion of external force would result in more obvious non-sphericity when the binding force is small. The erosion of the external force would also lead to sizable variations in $\mathrm{F_{33}/F_{11}}$, $\mathrm{F_{44}/F_{11}}$, $\mathrm{F_{12}/F_{11}}$, and $\mathrm{F_{34}/F_{11}}$. Specifically, the sign of $\mathrm{F_{33}/F_{11}}$, $\mathrm{F_{44}/F_{11}}$, $\mathrm{F_{12}/F_{11}}$, and $\mathrm{F_{34}/F_{11}}$ could be modified with the variation of $f$ in specific scattering angle ranges.

From the comparisons of Figures $6 - 8$, we can see how the aspect ratio of the original dust affect the impacts of $f$. Generally, the effects of $f$ on the scattering matrix are significantly affected by original aspect ratios. Modifying the aspect ratio, the $\mathrm{F_{33}/F_{11}}$, $\mathrm{F_{44}/F_{11}}$, $\mathrm{F_{12}/F_{11}}$ of dust with different $f$ could exhibit rather different trends with the scattering angles. From Figures $6 - 9$, we can see the comparison of the scattering matrix of dust with different binding forces (R). With an aspect ratio of 2:1, the phase function exhibits smaller backward scattering with increasing $f$ when R = 0, while the opposite

phenomenon was observed for R = 1. Besides, the $\mathrm{F_{22}/F_{11}}$ decreases with the increase of $f$ when R = 0, and the opposite phenomenon was also observed for R = 1. The finding can be explained from the following aspects. When R = 0, the binding force from the center of the dust is small, so the shape of the dust become more irregular with the increase of the erosion degree, and $\mathrm{F_{22}/F_{11}}$ becomes smaller. On the other hand, with a larger R, the large binding force would constrain the dust shape, and the dust becomes more spherical with the mass loss, so $\mathrm{F_{22}/F_{11}}$ become more close to 1.

#### 3.1.2  The applicability of spheroids


From Figures 4 - 9, we can also see the comparison of the phase matrix of dust with irregular shapes and best-fitted spheroids. As shown in Figure 4, when the particle size is small, the deviations between the scattering matrix of dust with irregular shapes





and those fitted using the spheroids are not substantial. Thus, the spheroid model can provide a reasonable estimation for small dust. However, we can see some small differences. With an original aspect ratio of 2:1 and an R of 0, the spheroid model would
underestimate the forward scattering and overestimate the backward scattering of $F_{11}$. Besides, the dust with irregular shapes can exhibit more obvious non-sphericities than the spheroids, so the $F_{22}/F_{11}$ of the dust with irregular shapes exhibits smaller values than those fitted using spheroids.

With a large particle size, the differences of the scattering matrix of dust with irregular shapes and spheroids become rather obvious. The best-fitted spheroids can generally reproduce the $F_{11}$ trend of dust with irregular shapes, while some obvious
differences are observed at the backward scattering angles. The dust with irregular shapes generally exhibit more obvious non-sphericity than the spheroids, so the $F_{22}/F_{11}$ values deviate more largely with 1 compared to those of spheroids. The trends for the $F_{33}/F_{11}$ and $F_{44}/F_{11}$ of dust with irregular shapes are similar to those of best-fitted spheroids. On the other hand, the trends of the $F_{12}/F_{11}$ and $F_{34}/F_{11}$ of dust with irregular shapes can be rather different from those of best-fitted spheroids, and the sign can be modified from negative to oppositive at some scattering angles if substituting the irregular dust with best-fitted
spheroids.

Figures 6 - 8 show similar results, but for different original aspect ratios. The original aspect ratio has a significant impact on the applicability of spheroids. With an original aspect ratio of 1:1, the spheroids fit the scattering matrix of irregular relatively well, while the fits of spheroids are relatively bad for the dust with an original aspect ratio of 2:1 and 1:2. The reason is that the mass of spherical particles is lost relatively uniformly, and the overall structure can be well represented by a spheroid.

Figure 6 and Figure 9 show how the binding force from the mass center affects the applicability of spheroids. As shown in Figure 8, when the binding force is small (R = 0), the scattering matrix differences between dust with irregular shapes and best-fitted dust are rather obvious as $f$ increases to 0.8. However, as R increases to 1, the difference turns much smaller when $f$ = 0.8. This can be explained from physical points. When the binding force is small, the mass of the dust is lost uniformly with the erosion, so the shapes can be much different from spheroids. However, with a large binding force, the mass loss is
constrained by the mass center, so the erosion is relatively uniform, and the shapes after erosion are close to spheroids.

Table 2 shows the scattering/extinction cross-sections of dust with irregular shapes. Fixing the aspect ratio to 2:1, with $f$ and $R$ varying, the variations of $C_{ext}$ and $C_{sca}$ are not substantial, and they are under 3%. However, the best-fitted $C_{ext}$ and $C_{sca}$ decrease substantially with the $f$ increasing. Fixing the aspect ratio to 2:1 and $R$ to 0, as $f$ increases from 0.1 to 0.8, the best-fitted $C_{ext}$ and $C_{sca}$ decrease by approximately 30%. When the $f$ is small, the deviation of $C_{ext}$ and $C_{sca}$ between the
irregular dust and best-fitted spheroids are not substantial, while the difference turns obvious as $f$ increases. When the aspect ratio is 2:1, $f$ = 0.8, $R$ = 0, and $d_p$ = 0.2 $\mu$m, the relative difference of $C_{ext}$ and $C_{sca}$ can reach approximately 30%. However, the deviations are mitigated when $R$ increases, as the large binding force would constraining the dust shape becoming more complex, and the retrieved aspect ratio is more close to 1:1.



**Table 2.** The scattering/extinction cross-section of dust with irregular shapes.

| Aspect Ration | $f$ | R | Diemeter ($\mu$m) | $C_{ext}$ ($\mu m^2$) | $C_{ext\_best\_fit}$ ($\mu m^2$) | $C_{sca}$ ($\mu m^2$) | $C_{sca\_best\_fit}$ ($\mu m^2$) | Fitted aspect ratio |
|---|---|---|---|---|---|---|---|---|
| 2:1 | 0.1 | 0 | 0.2 | 0.00622 | 0.00618 | 0.00579 | 0.00577 | 1.0 |
| 2:1 | 0.1 | 0 | 0.8 | 2.0189 | 1.98123 | 1.9714 | 1.93546 | 0.64 |
| 2:1 | 0.1 | 0 | 2.0 | 8.0645 | 7.94556 | 7.3306 | 7.20796 | 2.06 |
| 2:1 | 0.8 | 0 | 0.2 | 0.00606 | 0.00431 | 0.00562 | 0.00396 | 0.4 |
| 2:1 | 0.8 | 0 | 0.8 | 1.9282 | 1.72175 | 1.8818 | 1.68243 | 0.4 |
| 2:1 | 0.8 | 0 | 2.0 | 9.9786 | 9.71013 | 9.2322 | 8.96537 | 2.81 |
| 2:1 | 0.8 | 1 | 0.2 | 0.00620 | 0.00602 | 0.00579 | 0.00561 | 1.26 |
| 2:1 | 0.8 | 1 | 0.8 | 2.0790 | 2.0715 | 2.0333 | 2.0258 | 0.88 |
| 2:1 | 0.8 | 1 | 2.0 | 8.1955 | 8.2273 | 7.4695 | 7.5026 | 1.12 |

## 3.2 The skylight polarization of dust with irregular shapes

### 3.2.1 Effects of irregular shapes

To investigate the effects of dust shape on the polarized remote sensing signal, the normalized radiance (I), PBRF, and DoLP were calculated. Figures 10 - 12 show the effects of the erosion degree on the polarized remote sensing signal. In the plots, the backscattering direction is on the meridian plane with a zenith angle of 60° and a relative azimuth of 180°. As shown in Figure 10, the differences of normalized radiance (I), PBRF, and DoLP between the erosion fraction ($f$) of 0.1 and 0.8 are not substantial. Fixing $d_p$ to 0.2 $\mu$m, with $f$ increasing, the variation of I, PBRF, and DoLP are not substantial. Besides, the trends of I, PBRF, and DoLP with the relative azimuth angles and zenith angles are similar.

Nevertheless, with the particle size increasing, the erosion degree has more obvious impacts on the normalized radiance (I), PBRF, and DoLP. Figures 11 - 12 show similar results as Figure 10, but for dust with a $d_p$ of 0.8 and 2.0 $\mu$m, respectively. Different from dust with a $d_p$ of 0.2 $\mu$m, when the particle size increases to 0.8 and 2.0 $\mu$m, the erosion fraction has a significant impact on I, PBRF, and DoLP. The effects of $f$ are significantly related to the particle size. Fixing $d_p$ to 0.8 $\mu$m, when $f$ increases from 0.1 to 0.8, the normalized radiance exhibits a slightly decrease at backward scattering angles, and obvious increase is observed at forward scattering angles. The similar phenomenon was observed at $d_p$ of 2.0 $\mu$m.

With a $d_p$ of 0.8 $\mu$m, with increasing $f$, PBRF decreases at forward scattering angles but increase when the relative azimuth angle ranges from approximately 0° to 90° and the zenith angle is around 45°, and when both the relative azimuth angles and the zenith angle range from 60° to 90°. However, when $d_p$ is 2.0 $\mu$m, PBRF decreases when the relative azimuth angle is around 105° and the zenith angle is around 90°, and when the zenith angle is around 20° and the relative azimuth angle ranges from approximately 0° to 135°. Besides, with a $d_p$ of 2.0 $\mu$m, PBRF increases when the relative azimuth angle is around 60°





and the zenith angle is around 90°, and when the relative azimuth angle ranges from approximately 0° to 60° and the zenith angle is around 65°, which is rather different from the angular distribution of dust with a $d_p$ of 0.8 $\mu$m.

The effects of $f$ on DoLP are also significantly related to $d_p$. When $d_p$ is 0.8 $\mu$m, DoLP decreases at forward scattering angles but increases when the relative azimuth angle ranges from approximately 0° to 120° and the zenith angle is around 40°, and when both relative azimuth angles and zenith angle range from 60° to 90°. However, with a $d_p$ of 2.0 $\mu$m, when modifying $f$ from 0.1 to 0.8, a slight decrease in DoLP is found when the relative azimuth angle ranges from 0° to 60 ° and the zenith angle is around 30°. Besides, fixing $d_p$ to 2.0 $\mu$m, as $f$ increases from 0.1 to 0.8, an obvious increase in DoLP is observed

when the zenith angle ranges from 60 ° to 90 ° and the relative azimuth angle is around 60°. Also, DoLP increases when the zenith angle is around 60° and the relative azimuth angle ranges from approximately 0° to 60° with $f$ increasing from 0.1 to 0.8. Figures 10 - 12 also show that the polarized light signal is rather sensitive to the particle size, which agrees with the findings of previous studies.

    Figure 13 compares normalized radiance, PBRF, and DoLP of dust with different binding forces. With an original aspect

ratio of 2:1, as R increases from 0 to 1, the dust shape can become more spherical with the erosion. It could be seen from Figure 13 that I, PBRF, and DoLP are significantly affected by the binding force. Fixing the original aspect ratio to 2:1 and the particle diameter to 2.0 $\mu m$, with R increasing from 0 to 1, a slight decrease in the normalized radiance is observed at backward scattering angles, and an obvious increase in the normalized radiance is observed in the forward scattering angles. Modifying R from 0 to 1, PBRF shows an obvious increase at backward scattering angles, and a slight decrease was observed when both

relative azimuth angles and zenith angles are approximately 90°. As R varies from 0 to 1, DoLP increases significantly at forwarding scattering angles and decreases when relative azimuth angles range from 90° to 120° and zenith angles are 70° to 90°. Besides, DoLP also decreases when relative azimuth angles range from 0° to 90° and zenith angles are 30° to 50°. Thus, the angular distributions of normalized radiance, PBRF, and DoLP are significantly affected by the dust shape and particle size.

### 3.2.2   The deviations of the spheroid model

Figure 14 shows the difference of normalized I between the dust with irregular shapes and best-fitted spheroids. As both the particle size and erosion degree are small, the relative differences between the dust with irregular shapes and spheroids are not substantial, and the absolute value is below 3% when $f = 0.1$. However, even with small particle size, as $f$ increases to 0.8, the relative differences are rather more obvious, which range from approximately -6% to 5%. With a $d_p$ of 0.2 $\mu$m, the best-fitted spheroids underestimate the normalized radiance at backward scattering angles and overestimate the normalized radiance at

forwarding angles.

    As the particle diameter increases to 0.8 $\mu$m, the relative difference of normalized radiance between the dust with irregular shapes and spheroids becomes more obvious, and the relative difference in radiance can vary in the range of -10 to 10. Different from dust with a $d_p$ of 0.2 $\mu$m, when $d_p = 0.8$ $\mu$m, the deviations of the spheroid model are significantly affected by the erosion degree (i.e. $f$). When $f = 0.1$, the spheroid model underestimates the radiance at backward scattering angle but overestimates

the radiance when the zenith angles range from 10° to 85° and relative azimuth angles range from 0° to 120°. Nevertheless,





with an $f$ of 0.8, the spheroid model would overestimate the radiance at backward scattering angles, and underestimate the radiance at forwarding angles.

As $d_p$ further increases to 2.0 $\mu$m, the angular distribution of the radiance difference between the dust with irregular shapes and spheroids is further modified. Although the difference in radiance decreases compared to that for dust with a $d_p$ of 0.8 $\mu$m

when $f = 0.1$, as $f$ increase to 0.8, the absolute value of the relative difference in radiance can exceed 10%. Besides, the angular distributions of the difference also vary significantly with modifying $f$. Fixing $d_p$ to 2.0 $\mu$m, the spheroid model overestimates the radiance at forwarding scattering angles while underestimates the radiance at backward scattering angles when $f = 0.1$, and an opposite phenomenon is observed when $f = 0.8$.

The spheroid model can also provide inaccurate estimations for PBRF. As shown in Figure 15, with a small particle size ($d_p$

= 0.2 $\mu$m), the difference of PBRF between the dust with irregular shapes and spheroids is not substantial, and the maximum absolute value is below approximately 0.005. For dust with a $d_p$ of 0.2 $\mu$m, the spheroid model generally underestimates the PBRF at backward scattering angles while overestimating the PBRF at forwarding scattering angles. As $d_p$ increases to 0.8 $\mu$m, the PBRF differences between dust with irregular shapes and spheroids become rather obvious, and the difference varies from approximately -0.015 to 0.025. Different from dust with a $d_p$ of 0.2 $\mu$m, when $d_p$ increases to 0.8 $\mu$m, the spheroid model

generally overestimates the PBRF at backward scattering angles but overestimates the PBRF when the zenith angles range from 70° to 80° and relative azimuth angles range from approximately 0° to 30°. As $d_p$ further increases to 2.0 $\mu$m, the maximum absolute value of the PBRF difference shows decreases, and the difference varies in the range of approximately -0.015 to 0.015. when $f = 0.1$, the spheroid model generally overestimate the PBRF, but the spheroid model would underestimates the PBRF when the zenith angles are around 70° and relative azimuth angles range from 0° to 60°.

Figure 16 shows the comparison of DoLP between dust with irregular shapes and best-fitted spheroids. Similar to normalized radiance and PBRF, the DoLP differences between dust with irregular shapes and spheroids are not substantial when $d_p = 0.2$ $\mu$m, and the absolute value of the difference does not exceed 0.02. As $d_p$ increases to 0.8 $\mu$m, the difference between the spheroid model and dust with irregular shapes becomes rather obvious, and the difference varies from approximately -0.05 to 0.15. Besides, with a $d_p$ of 0.8 $\mu$m, the PBRF difference generally exhibits similar angular distributions for $f = 0.1$ and $f = 0.8$.

The spheroid model generally overestimates the DoLP at backward scattering angles. As $d_p$ increases to 2.0 $\mu$m, the difference decreases, and it varies from approximately -0.03 to 0.05. Besides, the angular distributions of DoLP are also different when $f$ varies. The spheroid model generally overestimates DoLP when $f = 0.1$, while some underestimations are found when the zenith angles are around 65° and relative azimuth angles range from approximately 0° to 30°.

Figure 17 compares the deviations of spheroids for estimating the normalized radiance, PBRF, and DoLP of dust with

different binding forces. The binding forces have a significant impact on the deviations. Fixing $d_p$ to 0.8 $\mu$m, original aspect ratio to 2:1, and $f$ to 0.8, with $R$ increasing from 0 to 1, the absolute values of deviations of radiance, PBRF, and DoLP decrease substantially. When $R = 0$, the relative difference of radiance vary from -6% to 5%, while varies in the range of approximately -1.5% to 1% when $R = 1$. As $R$ increases from 0 to 1, the PBRF differences change from the range of approximately -0.008 - 0.01 to approximately -0.0005 to 0.002, and the DoLP differences change from approximately -0.03 - 0.04 to approximately

-0.008 - 0.008. The physical points can explain why the difference decreases with R increasing. With larger $R$, the binding



force from the mass center increases, which can constrain the shape from becoming more complex, so the dust shape is close to the spheroid.

## 4  Summary and Conclusions

The spheroid model was commonly applied in the aerosol component retrievals based on the polarized light, while the applica-
bility of the spheroid model on estimating the polarization characteristics of dust with more complex shapes is still unclear. In the atmosphere, the dust shapes are various and a single model is difficult to represent the complex shapes of dust. To calculate the radiative properties of complex dust, we proposed a tunable model to represent dust with various shapes. We assumed that the dust shapes are mainly affected by two factors: (1) The dust shape can vary with the erosion of external force, which can lead to the loss of mass. (2) The binding force from the center of mass can prevent the loss of dust mass. We proposed
an algorithm with two tunable parameters to simulate the effects of these two factors, and various complex dust shapes were generated. As we used tunable parameters to represent various dust shapes, our model is helpful for the parameterization of the radiative properties of dust with different shapes. Besides, To evaluate the applicability of spheroids, the aspect ratio was retrieved using the first elements of the scattering matrix (i.e. phase function), and then the scattering properties of dust with irregular shapes and spheroids were compared.

The single scattering properties of dust with irregular shapes were investigated. We found that both the erosion of external force and binding force from the mass center can have a significant impact on the dust shapes, so significantly affect the single scattering properties of dust. When the particle size is small, the effects of dust shapes on the scattering matrix are relatively insensitive. However, with the particle size increasing, the dust shape can have rather obvious impacts on the scattering matrix, and even the sign of $F_{33}/F_{11}$, $F_{44}/F_{11}$, $F_{12}/F_{11}$, and $F_{34}/F_{11}$ could be modified with the variation of $f$ in specific scattering
angle ranges. The applicability of the best-fitted spheroids on estimating the scattering matrix was evaluated. The best-fitted spheroids can generally reproduce the $F_{11}$ of dust with irregular shapes, while the other elements show substantial differences. With a small particle size, the deviations of the scattering matrix between best-fitted spheroids are not substantial, while the deviations become substantial with the particle size increasing. Besides, the sign of $F_{12}/F_{11}$ and $F_{34}/F_{11}$ can be modified from negative to oppositive at some scattering angles if substituting the irregular dust with best-fitted spheroids. Our findings
also show that the binding force can affect the applicability of spheroids. Generally, with larger binding forces, the dust shapes are constrained from becoming more complex, and the spheroid model could provide relatively reasonable estimations. As the binding force is small, the deviation of extinction/scattering cross-section generally increases with the erosion degree, and the relative difference can reach approximately 30% when the erosion degree is large, while the deviation is mitigated with the binding force increasing. Besides, when increasing $R$, the retrieved aspect ratio is more close to 1:1.

To see how the dust shapes affect the polarimetric remote sensing, we have calculated the normalized radiance, PBRF, and DoLP of dust using the SOS model. Our findings show that dust shapes have a relatively unobvious impact on the normalized radiance, PBRF, and DoLP when the particle size is small, while the effects become rather obvious with the particle size increasing. Our findings show that both the erosion degree and the binding force can significantly affect the angular distribution



of normalized radiance, PBRF, and DoLP. The deviations of best-fitted spheroids were also investigated. When the particle size

is small, the spheroid model can provide good estimations. With a $d_p$ of 0.2 $\mu$m, the relative difference of normalized radiance does not exceed 3%, and the absolute values of the deviation of PBRF and DoLP are below 0.005 and 0.02, respectively. With the particle size increasing, the difference becomes much more substantial. The relative difference of normalized radiance can exceed 10%, and the deviation of PBRF and DoLP can vary in the range of -0.015 - 0.025 and the range of -0.05 to 0.15. Thus, the use of the spheroid model in the component retrievals based on the polarized light should consider the effects of more

complex dust shapes.

*Acknowledgements.* This work was financially supported by the National Outstanding Youth Foundation of China (Grant No. 41925019) and the National Natural Science Foundation of China (Grant No. 41871269). We particularly thank Dr. Michael Mischenko for making the T-matrix code publicly available.



*Author contributions.* JL and ZL designed the idea. JL developed the models, performed the computations, and wrote the paper. ZQL, CF,
HX, YZ, WH, LQ, HG, MZ, and YL verified results. ZL revised the paper and supervised the findings of this work. All authors discussed
the results and contributed to the final paper.

*Competing interests.* The authors declare that they have no conflict of interest.





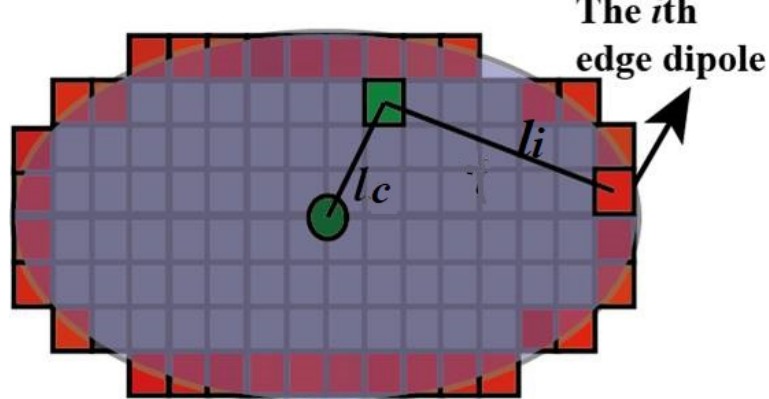

**Figure 1.** The generation of irregular dust.

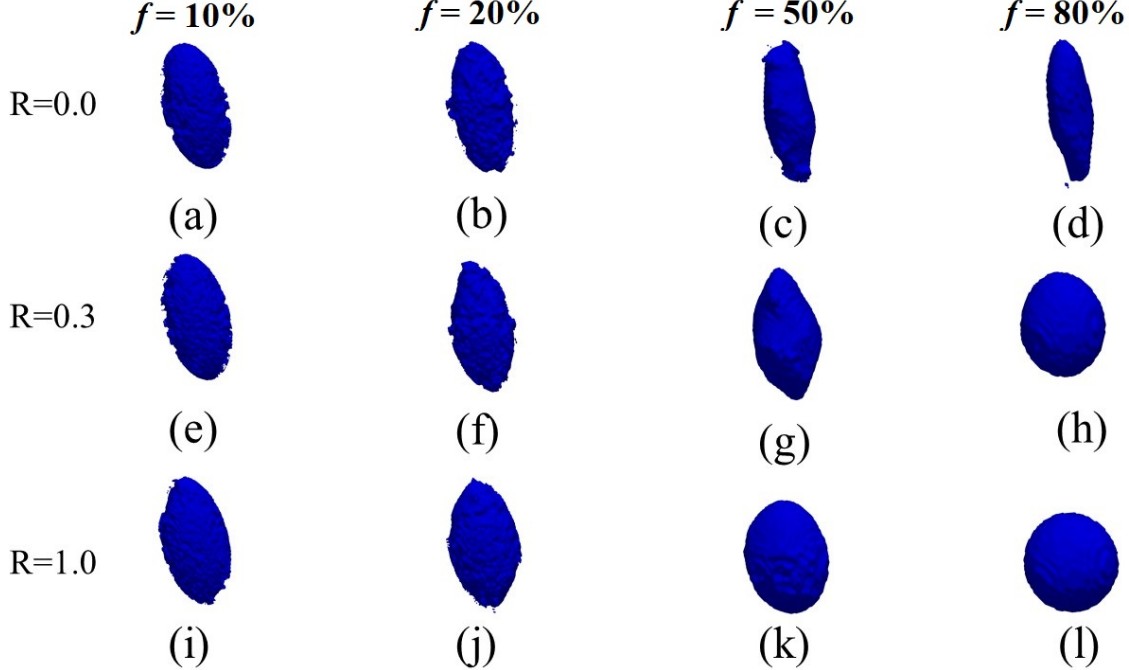

**Figure 2.** The typical morphologies of simulated dust.





**Figure 3.** The scattering matrix of spherical particles calculated using the DDSCAT and T-matrix codes, respectively, where $d_p = 0.4\ \mu$m.





**Figure 4.** The scattering matrix of dust with irregular shapes, where the aspect ratio is 2:1, $d_p$=0.2 $\mu$m.



**Figure 5.** Similar to Figure 4, but for $d_p$=0.8 $\mu$m.

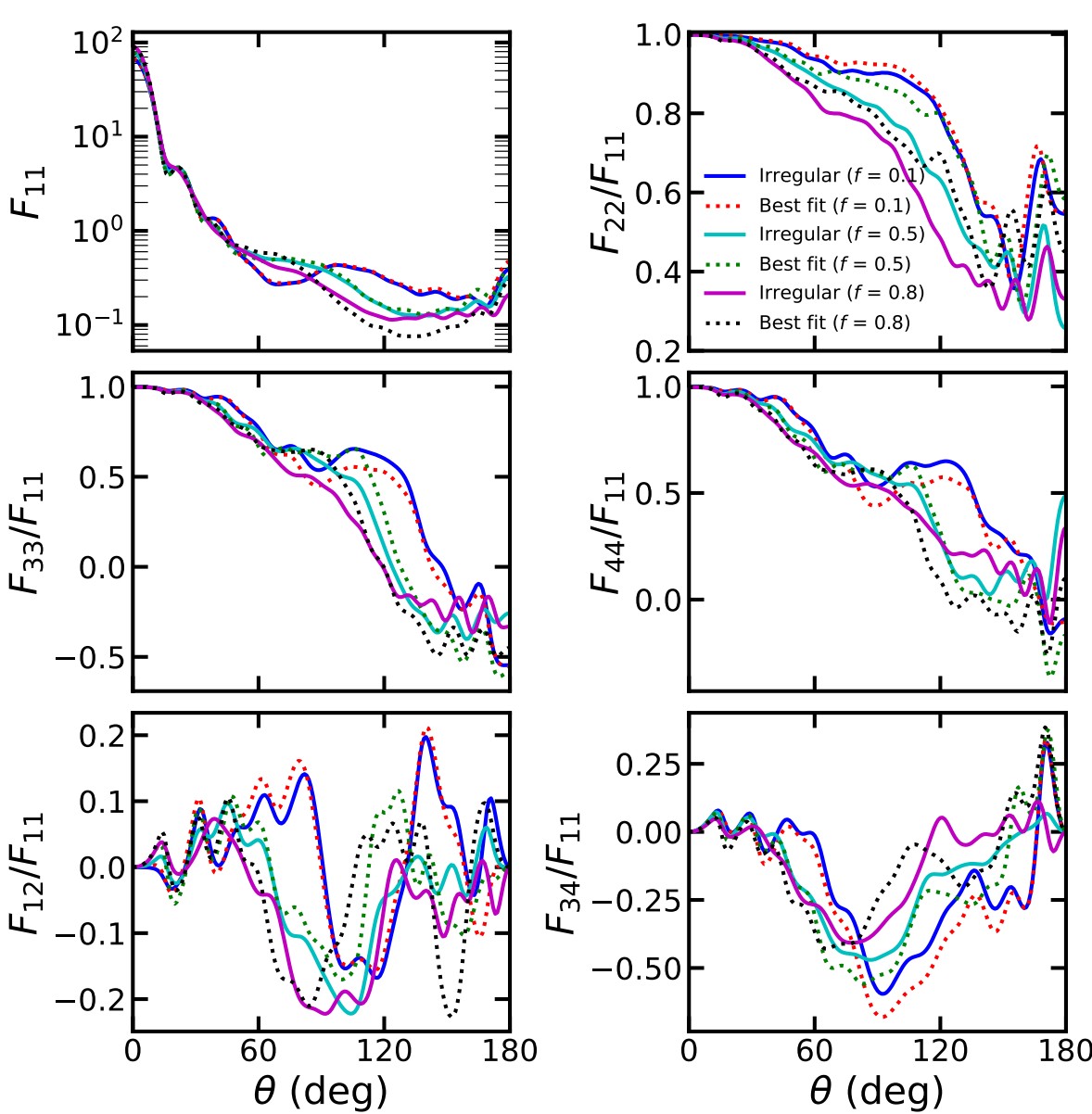

**Figure 6.** Similar to Figure 4, but for $d_p$=2.0 $\mu$m.





**Figure 7.** Similar to Figure 6, but for a aspect ratio of 1:1.







**Figure 8.** Similar to Figure 6, but for a aspect ratio of 1:2.





**Figure 9.** Similar to Figure 6, but for $R = 1$.



**Figure 10.** The polarimetric characteristics of dust with irregular shapes, where the aspect ratio is 2:1, $d_p = 0.2$ $\mu$m, $R = 0$.





**Figure 11.** Similar to Figure 10, but for $d_p$=0.8 $\mu$m.





**Figure 12.** Simialr to Figure 10, but for $d_p$=2.0 $\mu$m.

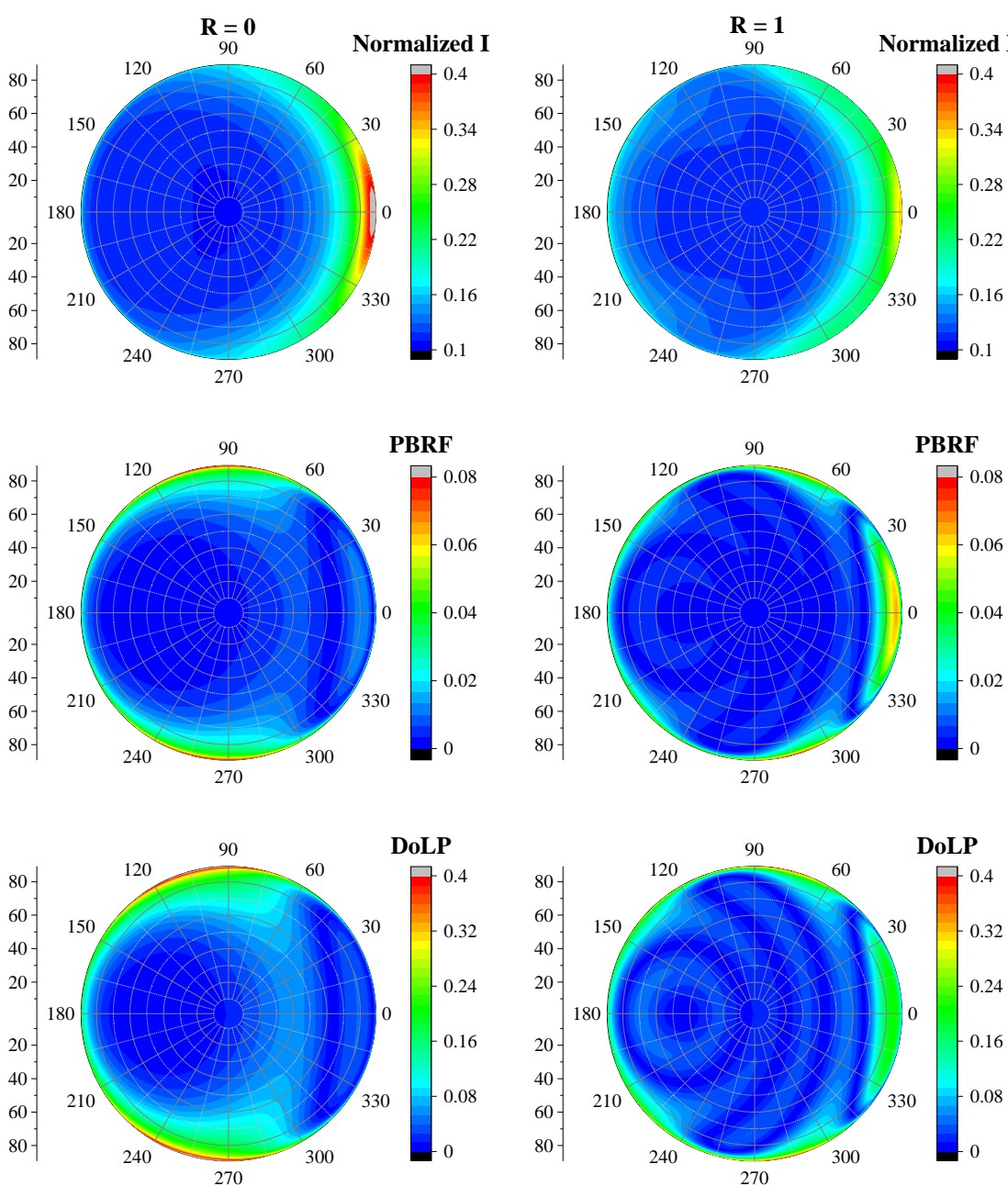

**Figure 13.** The polarimetric characteristics of dust with irregular shapes, where the aspect ratio is 2:1, $d_p = 2.0\ \mu$m, $f = 0.8$.





**Figure 14.** The relative difference of normalized radiance between dust with irregular shapes and spheroids, where the aspect ratio is 2:1, $R$ = 0.







**Figure 15.** Simialr to Figure 14, but for PBRF.





**Figure 16.** Simialr to Figure 14, but for DoLP.



**Figure 17.** The relative difference of normalized radiance between dust with irregular shapes and spheroids, where the aspect ratio is 2:1, $f$ = 0.8, $d_p$ = 0.8 $\mu$m.



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
