# Peer review of "The polarimetric characteristics of dust with irregular shapes: Evaluation of the spheroid model for single particles"

_Atmospheric Measurement Techniques, 2022_

## Referee Comment (RC1)

The paper "*The polarimetric characteristics of dust with irregular shapes: Evaluation of the spheroid model*" presents and discusses the development of a new dust shape model using DDA calculations. The authors simulate the effect of external forces (i.e. wind or rain) on initially perfectly shaped spheroid particles of dust nature, by removing dipoles close to the surface of the particle. Further, the effect of the binding force (the force from the particle centre of mass) is accounted for; and particles with larger binding force seem to turn to more spherical as the external force acts upon them. The authors further fit the resulting phase function of irregular shaped dust with spheroid particles (simulations for spheroids performed using the T-matrix algorithm) and compare the scattering matrix elements of irregular and spheroid particles. As a last step, radiative transfer simulations assuming both irregular shaped dust and the best-fitted spheroids are also performed and compared.

The study falls well within the scope of AMT and the results could be very significant for scientific community. Nevertheless, in order to help improving the manuscript, I would kindly suggest the authors to take into account the following specific comments.

 As I also stated in my initial review of the manuscript, I consider the range of input parameter values selected for the calculations, quite limited. The simulations are performed in only one specific wavelength (670nm) which is a frequently used wavelength for ground-based and satellite polarimetric measurements. AOD, SZA, surface albedo and complex refractive index (m) were also selected as single values.

For the latter, the authors select to use m = 1.52 + i0.005 for their calculations. However, the previous literature cited to justify this selection, corresponds to either dust mixtures with more absorbing particles (i.e. smoke; Dey at al., 2006) or results for dust have been omitted (Beelen et al. (2014). I realize that the main conclusions of the study won't change much, however I believe that additional values should be accounted for (see for example studies from Petzold et al. (2009) (k ranges from 0.0003 to 0.0017 at 700nm); Wagner et al. (2012) (k ranges from 0.0023 to 0.0051 at 655nm) or at least the authors should discuss possible effects of different refractive indices on their simulations.

In the following, I try to illustrate my concerns with a simple example where I have used the spheroid kernels developed and presented in Dubovik et al. (2006). Assuming a mono-modal, lognormal size distribution with geometric radius  $r_g = 2.32 \mu m$  with a standard deviation  $\sigma = 0.02$  (see Fig. 1; as narrow as possible SD to simulate as closely a single particle), the normalized scattering matrix elements for particles of an axial ratio of **2.07** and **0.53** are plotted in Fig.2 and Fig.3 respectively. The elements are calculated for real part of the refractive index n = 1.54 and 2 different imaginary refractive index (k) values:

- $m_1 = 1.54 + i0.006$  (green lines) which is more close to the value selected by the authors
- $m_2 = 1.54 + i0.0008$  (purple lines)

As it can be seen from figures 2 and 3, the effect of k on the scattering matrix elements can –at certain angles- cause a relative difference  $(m_1 - m_2/m_1)$  of up to 60% for the phase function, 20% for  $P_{12}/P_{11}$  and even higher for  $P_{22}/P_{11}$  for an axial ratio of 2.07 and similar for 0.53.

Fig.1) Mono-modal, lognormal size distribution used in the simulations shown in Fig.2 and Fig. 3

Fig.2) Scattering matrix elements for spheroids of axial ratio 2.07, the size distribution shown in Fig.1 and complex refractive index m = 1.54 + i0.0008 (purple lines)/i0.006 (green lines).

---

## Author Comment (AC1)

**Response to the comments of Reviewer #2**

First of all, we would like to thank the two anonymous reviewers for their thoughtful reviews and valuable comments on the manuscript. In the revision, we have accommodated all the suggested changes into consideration and revised the manuscript accordingly. All changes are highlighted in the revised manuscript in **BLUE** in the revision. In this response, the questions and comments of reviewers are in **BLACK** font, and responses are highlighted in **BLUE.** The changes made in the revised manuscript are marked in **RED** font.

**Comments:** The paper "*The polarimetric characteristics of dust with irregular shapes: Evaluation of the spheroid model*" presents and discusses the development of a new dust shape model using DDA calculations. The authors simulate the effect of external forces (i.e. wind or rain) on initially perfectly shaped spheroid particles of dust nature, by removing dipoles close to the surface of the particle. Further, the effect of the binding force (the force from the particle centre of mass) is accounted for; and particles with larger binding force seem to turn to more spherical as the external force acts upon them. The authors further fit the resulting phase function of irregular shaped dust with spheroid particles (simulations for spheroids performed using the T-matrix algorithm) and compare the scattering matrix elements of irregular and spheroid particles. As a last step, radiative transfer simulations assuming both irregular shaped dust and the best-fitted spheroids are also performed and compared.

The study falls well within the scope of AMT and the results could be very significant for scientific community. Nevertheless, in order to help improving the manuscript, I would kindly suggest the authors to take into account the following specific comments.

**Response:** Thanks for your comments. The responses are shown in the following.

**Comments:** 1) As I also stated in my initial review of the manuscript, I consider the range of input parameter values selected for the calculations, quite limited. The simulations are performed in only one specific wavelength (670nm) which is a frequently used wavelength for ground-based and satellite polarimetric measurements. AOD, SZA, surface albedo and complex refractive index (m) were also selected as single values.

**Response:** Thanks for your comments. We selected 670 nm wavelength as a typical example to show the effects of irregular shape on the polarimetric characteristics because it is a frequently used wavelength for ground-based and satellite polarimetric measurements. For sensitivity analysis, we have added some cases in the 490nm and 865 nm in the revised manuscript. As you said, the AOD, SZA, surface albedo and complex refractive index (m) can have an important impact on polarimetric characteristics.    In the revised manuscript, we added some sensitivity analysis to investigate how these parameters affect the effects of dust shapes. However, we think that these materials don't affect the main conclusion of this manuscript.

Thus, we put these materials in the support information.

**Comments:** For the latter, the authors select to use m = 1.52 + i0.005 for their calculations. However, the previous literature cited to justify this selection, corresponds to either dust mixtures with more absorbing particles (i.e. smoke; Dey at al., 2006) or results for dust have been omitted (Beelen et al. (2014). I realize that the main conclusions of the study won't change much, however I believe that additional values should be accounted for (see for example studies from Petzold et al. (2009) (k ranges from 0.0003 to 0.0017 at 700nm); Wagner et al. (2012) (k ranges from 0.0023 to 0.0051 at 655nm) or at least the authors should discuss possible effects of different refractive indices on their simulations.

In the following, I try to illustrate my concerns with a simple example where I have used the spheroid kernels developed and presented in Dubovik et al. (2006). Assuming a mono-modal, lognormal size distribution with geometric radius rg = 2.32µm with a standard deviation σ = 0.02 (see Fig. 1; as narrow as possible SD to simulate as closely a single particle), the normalized scattering matrix elements for particles of an axial ratio of 2.07 and 0.53 are plotted in Fig.2 and Fig.3 respectively. The elements are calculated for real part of the refractive index n = 1.54 and 2 different imaginary refractive index (k) values:

- m1 = 1.54 + i0.006 (green lines) which is more close to the value selected by the authors
- m2 = 1.54 + i0.0008 (purple lines)

As it can be seen from figures 2 and 3, the effect of k on the scattering matrix elements can –at certain angles- cause a relative difference (m1 – m2/m1) of up to 60% for the phase function, 20% for P12/P11 and even higher for P22/P11 for an axial ratio of 2.07 and similar for 0.53.

**Response:** Thanks for your comments. In the revised manuscript, we have added sensitivity analysis for the refractive index. Three other refractive indices are considered:

- m1 = 1.52 - i0.0007
- m2 = 1.52- i0.0014

- m2 = 1.52-i0.01

Figure 1 shows the effects of refractive indices on the scattering matrix. Similar phenomenon was found as the Figures you presented, larger k can lead to smaller $F_{11}$ at backscattering angles. Besides, larger $F_{22}/F_{11}$ values were observed for larger k. The imaginary part of refractive index has non-negligible impacts on the scattering matrix. The relative deviations in $F_{11}$ between cases of m = 1.52-0.01i and m = 1.52 – 0.0007i can also exceed 60% at backward angles.

[Figure]

Figure 1 The sensitivity of scattering matrix of dust to the refractive index

Figure 2 shows the effects of k on the differences of scattering matrices between irregular dust and best-fitted spheroids. As shown in Figure 2, even though the angle distributions of the deviations between the irregular dust and best-fitted spheroids are similar for different k, some sizable differences for the deviations between irregular dust and best-fitted spheroids are observed at certain angles for different k. The differences of relative $\Delta$F11 between m = 1.52-0.01i and m =1.52 – 0.0007i can reach approximately 18% at backward scattering angles.

[Figure]

Figure 2 The differences of scattering matrices between irregular dust and best-fitted spheroids

We have also compared the polarimetric characteristics of dust with different aerosol optical depth (AOD), surface albedo, and imaginary parts of refractive indices. As shown in Figure 3 - 5 and Figure S14 – F16, the polarimetric characteristics of dust with irregular shapes share similar angular distributions for different AOD, surface albedo, and imaginary parts of the refractive index, and the modifications of AOD, surface albedo, and imaginary parts of the refractive index should not modify the main conclusions.

[Figure]

Figure 3 The polarimetric characteristics of dust with irregular shapes for different AOD, where the aspect ratio is 2:1, $d_p = 2.0$ $\mu m$, $f = 0.8$.

[Figure]

Figure 4 The polarimetric characteristics of dust with irregular shapes for different surface albedo, where the aspect ratio is 2:1, $d_p = 2.0$ $\mu m$, $f = 0.8$

[Figure]

Figure 5 The polarimetric characteristics of dust with irregular shapes for different k, where the aspect ratio is 2:1, $d_p$ = 2.0 μm, $f$ = 0.5.

**Comments:** Calculations in the manuscript assume single particles both for irregular dust and the best-fitted spheroids. Although there is no doubt that more realistic representations of dust particle shapes are needed, I wonder what happens if the authors assume randomly oriented ensemble of such irregular particles, and average their properties over a size distribution. I would expect that the characteristics of the irregular shapes smooth out. How are your results compared to those assuming poly-dispersed spheroids?

**Response:** Thanks very much for your comments. In this work, only single particles were considered as a first step toward exploring the applicability of spheroidal shapes. The bulk optical properties are not considered in this work due to the expensive computation costs. However, we have considered three dust particles which represent small, medium, and large particles for dust in the fine mode. The applicability of ensembles of spheroidal particles should be further investigated in the future, and this is a drawback of this work. We have added some clarifications for the drawbacks. In the future, the optical properties of irregular dust with a realistic size distribution would be considered and they would be applied in the polarimetric remote sensing. We have added some descriptions in the revised manuscript to illustrate the drawbacks of this paper and the future works that need to be conducted in order to apply our models in polarimetric remote sensing.

**Comments:** Phrasing needs significant improvements throughout the manuscript to make it more easy to follow.

Some specific examples are provided below:

3) Page 2, line 25: "Dust can also modify the cloud properties by serving as the cloud condensation nucleus (CNN), so play an indirect effect on the climate"

Consider rephrasing to something like: "Dust particles can also indirectly affect the Earth's climate, as they can serve as highly effective cloud condensation and ice nuclei (CCN and IN) and thus modify cloud lifetimes, albedo and microphysical properties"

**Response:** Thanks very much for your comments. We have corrected it in the revised manuscript.

4) Page 2, line 30: "Ground-based remote sensing and satellite remote sensing are the main techniques to retrieve aerosols"

-//- : "Ground-based and satellite measurements are the main remote sensing techniques to derive aerosol particle properties"

**Response:** Thanks very much for your comments. We have corrected it in the revised manuscript.

5) Page 2, line 37: "mainly derive the whole floor of aerosols"

-//- : "Mainly derive the aerosol properties through the total atmospheric column along with surface characteristics"

**Response:** Thanks very much for your comments. We have corrected it in the revised manuscript.

6) Page 2, line 43: "The extinction coefficient"

Maybe the authors here mean the ensemble averaged extinction cross section?

**Response:** Thanks very much for your comments. We have corrected it in the revised manuscript.

7) Page 4, line 94: "Under the erosion of the external forces, the mass of the dust would be lost. On the other hand, the binding force could constrain the loss of dust mass"

-//- : "Due to erosion forces acting on the particles, part of dust mass would be lost in the form of dust granules leaving the particle surface. However, binding force from the particle centre of mass could constrain this loss"

**Response:** Thanks very much for your comments. We have corrected it in the revised manuscript.

8) Page 4, line 113: "V0 denotes the volume lost in the erosion process" V0 here should be

replaced with Vlost.

**Response:** Thanks very much for your comments. We have corrected it in the revised manuscript.

9) Page 4, line 114: "the dust shapes are easier becomes spherical due to larger binding force"
-//- : "Dust particles eroded under external forces are easier to become more spherical when the binding force is large"

**Response:** Thanks very much for your comments. We have corrected it in the revised manuscript.

10) Page 5, line 120: "To reflect the Stokes vector of polarization, the normalized Stokes scattering matrix has six independent elements"
-//- : "For rotationally symmetric, randomly oriented particles, the normalized Stokes scattering matrix has six independent elements"

**Response:** Thanks very much for your comments. We have corrected it in the revised manuscript.

11) Page 7, line 160: "The scattering matrices of dust with different shapes are shown in Figures 4 – 6"
-//- : "The scattering matrices of dust with different irregular shapes and the corresponding spheroids that best fit the phase function are shown in Figures 4 – 6"

**Response:** Thanks very much for your comments. We have corrected it in the revised manuscript.

**Supporting Information for "The polarimetric characteristics of dust with irregular shapes: Evaluation of the spheroid model"**

Jie Luo[1], Zhengqiang Li[1,2], Cheng Fan[1], Hua Xu[1,2], Ying Zhang[1,2], Weizhen Hou[1,2], Lili Qie[1], Haoran Gu[1,2,3], Mengyao Zhu[1,2,4], Yinna Li[1,2], and Kaitao Li[1]

[1]State Environment Protection Key Laboratory of Satellite Remote Sensing, Aerospace Information Research Institute, Chinese Academy of Sciences, Beijing 100101, China
[2]University of Chinese Academy of Sciences, Beijing 100049, China
[3]College of Geography and Tourism, Anhui Normal University, Wuhu 241003, China
[4]College of Geoexploration Science and Technology, Jilin University, Changchun 130026, China

**Correspondence:** Zhengqiang Li (lizq@radi.ac.cn)

[Figure]

**Figure S1.** The scattering matrix of spherical particles calculated using the DDSCAT and T-matrix codes, respectively, where $d_p = 0.4 \ \mu$m.

[Figure]

**Figure S2.** The scattering matrix of dust with irregular shapes, where the aspect ratio is 2:1, $d_p$=0.8 $\mu$m.

[Figure]

**Figure S3.** The differnces of scattering matrix bettween the best-fitted spheroids and dust with irregular shapes, where the aspect ratio is 2:1, $d_p$=0.2 $\mu$m.

[Figure]

**Figure S4.** Similar to Figure S3, but for $d_p$=0.8 $\mu$m.

[Figure]

**Figure S5.** Similar to Figure S3, but for $d_p$=2.0 $\mu$m.

[Figure]

**Figure S6.** Similar to Figure S5, but for a aspect ratio of 1:1.

[Figure]

**Figure S7.** Similar to Figure S5, but for a aspect ratio of 1:2.

[Figure]

**Figure S8.** Similar to Figure S5, but for R = 1.

[Figure]

**Figure S9.** Similar to Figure S5, but for different imaginary parts of dust refractive indices (k), where *f* = 0.5.

[Figure]

**Figure S10.** The scattering matrix of dust with irregular shapes at different wavelenths, where the aspect ratio is 2:1, $d_p$=2.0 $\mu$m, m = 1.52 - 0.0014i.

[Figure]

**Figure S11.** The polarimetric characteristics of dust with irregular shapes for different AOD, where the aspect ratio is 2:1, $d_p = 2.0$ $\mu$m, $f =$ 0.8.

[Figure]

**Figure S12.** The polarimetric characteristics of dust with irregular shapes for different surface albedo, where the aspect ratio is 2:1, $d_p = 2.0$ $\mu$m, $f = 0.8$.

[Figure]

**Figure S13.** The polarimetric characteristics of dust with irregular shapes for different k, where the aspect ratio is 2:1, $d_p = 2.0$ $\mu$m, $f = 0.5$.

[Figure]

**Figure S14.** The difference of polarimetric characteristics between dust with irregular shapes and spheriods for different AOD, where the aspect ratio is 2:1, $d_p$ = 2.0 $\mu$m, $f$ = 0.8.

[Figure]

**Figure S15.** The difference of polarimetric characteristics between dust with irregular shapes and spheriods for different surface albedo, where the aspect ratio is 2:1, $d_p = 2.0 \ \mu$m, $f = 0.8$.

[Figure]

**Figure S16.** The defference of polarimetric characteristics between dust with irregular shapes and spheriods for different k, where the aspect ratio is 2:1, $d_p$ = 2.0 $\mu$m, $f$ = 0.5.

---

## Author Comment (AC2)

**Response to the comments of Reviewer #1**

First of all, we would like to thank the two anonymous reviewers for their thoughtful reviews and valuable comments on the manuscript. In the revision, we have accommodated all the suggested changes into consideration and revised the manuscript accordingly. All changes are highlighted in the revised manuscript in **BLUE** in the revision. In this response, the questions and comments of reviewers are in **BLACK** font, and responses are highlighted in **BLUE**. The changes made in the revised manuscript are marked in **RED** font.

**Comments:** The manuscript on "The polarimetric characteristics of dust with irregular shapes: Evaluation of the spheroid model" discusses the applicability of the spheroidal shape for reproducing the scattering properties of irregularly-shaped dust particles. This is a valuable study that is long-awaited by the community, thus it is of value to be published. That being said, the study is far from providing a complete answer on the applicability of spheroids for reproducing the scattering properties of dust, and it should be clearly presented as a first step towards this direction. In this context, the following limitations have to be highlighted and discussed:

**Response: Thanks for your comments. The responses are shown in follows.**

**Comments:** The study presents the scattering properties of single particles. As shown in several studies in the literature, with one of the most prominent the work of Dubovik et al. (2006), in order to reproduce the scattering properties of dust, ensembles of spheroidal particles are used (i.e. with a distribution of sizes and aspect ratios) and not single spheroidal particles. Thus, differences seen in the scattering of single spheroidal particles with single irregularly-shaped particles do not necessarily indicate the inability of an ensemble of spheroidal particles. These differences may be viewed as a first step towards exploring the applicability of spheroidal shapes for reproducing the scattering properties of dust, and this is how it should be presented in the manuscript. This limitation should be clearly highlighted, both in the abstract and in the rest of the manuscript, but also in the title, by changing it to: "The polarimetric characteristics of dust with irregular shapes: Evaluation of the spheroid model for single particles".

**Response:** Thanks for your comments. This is a really valuable suggestion. In this work, only single particles were considered as a first step towards exploring the applicability of spheroidal shapes. The applicability of ensembles of spheroidal particles should further investigated in the future, and this is a drawback of this work. We have added some clarifications for the drawbacks.

**Comments:** The sizes of dust particles in the study are quite limited, considering mainly the fine dust particles (i.e., radius up to  $2.0 \mu m$ ). This should be highlighted in the abstract.

**Response: Thanks for your comments. We have clarified it in the revised manuscript.**

**Comments:** The radiative transfer calculations presented in the manuscript use the scattering properties of single particles (if I understood correctly). This is unrealistic for the radiative transfer calculations in the atmosphere. Thus, the radiative transfer calculations presented in the manuscript should be re-calculated, using the scattering properties of ensembles of particles (not of single particles), otherwise this part (i.e. Sect. 2.4 and 3.2) should be omitted.

**Response:** Thanks for your comments. In this work, only single particles were considered, and the radiative transfer calculations are indeed unrealistic. However, the aims of this work is to investigate how the shapes of dust with different sizes affect the applicability of spheroids. Thus, the radiative transfer calculations could provide theoretical implications for the applicability of spheroids in calculating the polarized radiation. Thus, we don't delete the radiative transfer calculations would be conducted.

**Comments:** The quantification of the differences in the scattering properties of single spheroidal particles and single irregularly-shaped particles, is not thoroughly provided in the manuscript. Be specific and provide numbers.

**Response:** Thanks for your comments. We have quantified the differences in the revised manuscript.

**Comments:** As mentioned in the technical review, the use of English language in the manuscript is not optimum, and it should be thoroughly revised, especially for Sect. 1 (Introduction).

**Response:** Thanks for your comments. We have re-checked and revised the English carefully, and the modifications are marked in the revised manuscript.

General comments:

**Comments:** - Include in the abstract the limitations discussed in comments (1), (2) and (3) above.

Response: Thanks for your comments. We have included the limitations in the abstract:

"In this work, only the optical properties of single particles were considered. In the future, the applicability of an ensemble of spheroidal particles for reproducing the scattering properties and polarimetric characteristics of an ensemble of irregularly-shaped dust particles should be further investigated."

More specific comments:

Comments: Line 9 "... substantial deviations...": Provide quantification.

**Response:** Thanks for your comments. We have provided quantifications in the revised manuscript:

"The F11 relative differences of approximately 100% could be observed in some certain scattering angles. The maximum differences of other elements between irregular dust particles and best-fitted spheroids can reach approximately 0.3 - 0.8. Besides, the sign of F12/F11, F33/F11, F34/F11 and F44/F11 can be modified from negative to oppositive at some scattering angles if substituting the irregular dust with best-fitted spheroids."

Comments: Line 9 ".... is relatively large.": Provide the size (and size parameter) range used.

**Response:** Thanks for your comments. We have provided the size range in the revised manuscript.

**Comments:** Lines 14-15 "The deviations of the spheroid model... (VRT) model": Re-calculate the RT using ensembles of particles, or exclude this (see comment (3) above).

**Response:** Thanks for your comments. In this work, only single particles were considered, and the radiative transfer calculations are indeed unrealistic. However, this work aims to investigate how the shapes of dust with different sizes affect the applicability of spheroids. Thus, the radiative transfer calculations could provide theoretical implications for the applicability of spheroids in calculating the polarized radiation. Thus, we don't delete the radiative transfer calculations would be conducted.

**Comments:** Lines 20-21 "Thus, the use... dust shapes.": Rephrase this, based on comment (1) above.

**Response:** Thanks for your comments. We have rewritten this sentence as:

"Thus, the single spheroid model may lead to non-negligible deviations for estimating the polarimetric characteristics of single dust particles with more complex shapes. In the future, the applicability of an ensemble of spheroidal particles for reproducing the scattering properties and polarimetric characteristics of an ensemble of irregularly-shaped dust particles should be further investigated."

**Comments:** Lines 35-38 "However... of the surface": Rephrase (not optimum use of English). Also, the satellite aerosol retrieval algorithms do not only suffer from the "perturbs of the surface". Discuss and provide references.

Response: Thanks for your comments. We have rewritten this sentence as:

"However, satellite remote sensing may provide inaccurate estimates due to the poor understanding of aerosol optical properties. The traditional satellite aerosol retrieval algorithms mainly derive the entire aerosol layer based on radiation fluxes, while due to surface perturbations it is difficult to estimate the contribution of different components."

**Comments:** Lines 39-40 "The polarization... Li et al., 2016": Rephrase (not optimum use of English).

**Response:** Thanks for your comments. We have revised the sentence as:

"Polarization is more sensitive to the atmosphere and less disturbed by surfaces than radiation"

**Comments:** Line 47 "However, ... calculations,": Substitute it with "However, the full calculations require to use the vector radiative transfer,"

Response: Thanks for your comments. We have revised it in the revised manuscript.

Comments: Line 48 "In most remote sensing algorithms...": Start a new paragraph.

Response: Thanks for your comments. We have revised it in the revised manuscript.

Comments: Line 52 "Dust particles... 2011).": Delete this sentence.

Response: Thanks for your comments. We have revised it in the revised manuscript.

Comments: Line 53 "The spheroid model...": Do not start a new paragraph here.

Response: Thanks for your comments. We have revised it in the revised manuscript.

**Comments:** Line 54 "(Dubovik et al., 2006; 2011)": Substitute it with "(e.g., Dubovik et al., 2006)".

**Response:** Thanks for your comments. We have revised it in the revised manuscript.

**Comments:** Line 54 "Compared ... determined": Substitute it with "Compared to the spherical model, the aspect ratio of the particle needs to be determined.".

Response: Thanks for your comments. We have revised it in the revised manuscript.

Comments: Lines 54-55 "The retrieval... 2011).": Delete this sentence.

Response: Thanks for your comments. We have revised it in the revised manuscript.

**Comments:** Lines 58-68: Re-write the whole paragraph, based on the following: a) Spheroids are used as a model for reproducing the optical properties of dust, thus their aspect ratio is not necessarily a microphysical property of the particles, and it is usually not retrieved. E.g. Mishchenko et al. (1997) do not retrieve the aspect ratio of the spheroids. Delete the sentence "In traditional... of spheroids." b) The phase function of dust particles is reproduced using ensembles of spheroidal particles. There is no extensive study on the ability of the ensembles of spheroids to reproduce all the elements of the scattering matrix of dust. Emphasize that the current study is a first step towards this direction, showing the reproduction of the scattering matrix of single dust particles, using single spheroidal particles.

Response: Thanks for your comments. We have rewritten the paragraph as:

"Mishchenko et al. (1997) have used the spheroids to model the phase function of dust. Even though dust particles are not perfect spheroids, the spheroids can provide reasonable estimations for the phase functions with a wide aspect ratio distribution(Mishchenko et al., 1997). Kocifaj et al. (2008) have retrieved the aspect ratio based on the phase function and extinction coefficient simultaneously. Li et al. (2019) have shown that the polarimetric characteristics calculated assuming the microscope-measured aspect ratio is distinct from that using the inversion-based aspect ratios. Nevertheless, Spheroids are used as a model for reproducing the optical properties of dust, thus their aspect ratio is not necessarily a microphysical property of the particles. It is still unclear whether the spheroids that best fit the phase function can represent other elements of the scattering matrix and the polarization. Thus, it is desirable to know the applicability of spheroids on reproducing the other elements of the scattering matrix and the polarization."

Besides, at the end of the introduction, we have added some clarifications for the drawbacks of this work:

"In principle, in order to reproduce the scattering properties of dust, ensembles of spheroidal particles should be used (i.e. with a distribution of sizes and aspect ratios). However, as a first step towards exploring the applicability of spheroidal shapes for reproducing the scattering properties of dust, we just consider single particles in this work, and further investigations for ensembles of dust particles would be investigated in the future."

**Comments:** Line 72 "...Kahnert, 2015).": Include also "Gasteiger et al., 2011" (Gasteiger, J., Wiegner, M., Groß, S., Freudenthaler, V., Toledano, C., Tesche, M., and Kandler, K.: Modelling lidar-relevant optical properties of complex mineral dust aerosols, Tellus, B 63, 725–741, doi:10.1111/j.1600-0889.2011.00559.x, 2011.).

Response: Thanks for your comments. We have added the reference in the revised manuscript.

**Comments:** Lines 69-78 "Some modelling works... investigated.": Re-write the whole paragraph, based on comment (1) above.

**Response:** Thanks for your comments. We have rewritten the whole paragraph:

"Some modeling works have been conducted to investigate the optical properties of more irregular dust, and they have shown that the optical properties of dust are significantly affected by their shapes (Yang et al., 2007; Lindqvist et al., 2014; Bi et al., 2010; Liu et al., 2013; Escobar-Cerezo et al., 2017; Zubko, 2013; Kanngießer and Kahnert, 2021; Kemppinen et al., 2015; Kahnert, 2015; Gasteiger et al., 2011). However, in applications, we are more curious about whether the optical properties can be reproduced by a simplified model. Even though an ensemble of irregularly-shaped dust particles exist in the atmospheres, the optical properties may be reproduced by an ensemble of spheroidal particles with a wide aspect ratio distribution. As a first step, we are mainly focused on answering the following questions:

- Could the single spheroid with a best ftted aspect ratio reproduce the single-scattering properties of single dust with more complex shapes?

- Could the single spheroid with a best-fitted aspect ratio reproduce the polarimetric characteristics of single dust with more complex shapes?

- How do the dust shapes affect the scattering properties and polarimetric characteristics?

In the atmosphere, the dust shapes are various, and a single model is difficult to represent the complex shapes of dust, so we need to develop dust models which can represent various shapes. In principle, in order to reproduce the scattering properties of dust, ensembles of spheroidal particles should be used (i.e. with a distribution of sizes and aspect ratios). However, as a first step towards exploring the applicability of spheroidal shapes for reproducing the scattering properties of dust, we just consider single particles in this work, and further investigations for ensembles of dust particles would be investigated in the future. To answer the above questions, we proposed an irregular model to represent the dust with various morphologies, and the scattering properties were calculated using discrete dipole approximation (DDA) methods. Then, we retrieved the aspect ratio that best fits the phase function of dust with complex morphologies using the spheroid model, and the phase matrices of dust with complex morphologies and best-fitted spheroids were compared. Besides, the radiance and polarization were calculated using a vector radiative transfer (VRT) code based on plane-parallel successive-order-of-scattering (SOS), and the capabilities of spheroids for representing the radiance and polarization of irregular dust were evaluated."

**Comments:** Line 83: Before this line, start a new paragraph, discussing the limitations of the current work (see comments (1) and (2)). Describe this work as a first step towards a more complete analysis.

**Response:** Thanks for your comments. We have added some descriptions in the revised manuscript:

"In principle, in order to reproduce the scattering properties of dust, ensembles of spheroidal particles should be used (i.e. with a distribution of sizes and aspect ratios). However, as a first step towards exploring the applicability of spheroidal shapes for reproducing the scattering properties of dust, we just consider single particles in this work, and further investigations for ensembles of dust particles would be investigated in the future."

**Comments:** Line 90, Sect. 2.1: Mention (in this section) the lack of faceted particles in the analysis. See for example particles "E" and "F" in Fig. 1 in Gasteiger et al. (2011).

**Response:** Thanks for your comments. We have mentioned it in the revised manuscript:

"In this work, to evaluate the applicability of spheroids, the ideal shapes are assumed as spheroids. However, in the atmosphere, faceted dust particles were also observed in the atmosphere (Gasteigeret al., 2011), and these particles should also be investigated in the future."

**Comments:** Lines 103-104 "...and it can simulate the roughness of the surface.": The roughness of the surface is probably at smaller scales than the dipole size. Please discuss and provide references.

Response: Thanks for your comments. Some previous studies have also simulated the roughness of the surface using the DDA method. We have added some discussions in the revised manuscript:

"Previous studies have simulated the surface roughness by randomly adding or subtracting dipoles in random surface locations (Kemppinen et al., 2015b; Veghte et al., 2015). We adopted similar methods in this work. We used a random parameter  $R_n$ , which represents the randomness of external force, to simulate the roughness of the surface.  $R_n$  varies from 0 to 1. However, the roughness of the dust surface is also probably at smaller scales than the dipole size, which was not considered in this work."

**Comments:** Lines 113-114 "...and Vo denotes the volume lost in the erosion process. As shown in Figure 2, with a larger R, the dust shapes are easier becomes spherical due to larger binding force.": Substitute with "...and VLost denotes the volume lost in the erosion process. As shown in Figure 2, with a larger R, the dust shapes become more spherical due to the larger binding force."

**Response:** Thanks for your comments. We have revised it in the revised manuscript.

**Comments:** Lines 118-121 "The normalized... Mishchenko et al., 2002)": Substitute with "The normalized scattering matrix, extinction cross-section (Cext), and scattering cross-section (Csca) are the key parameters of the single scattering properties of aerosols (Mishchenko et al., 2002; Liu and Mishchenko, 2005). The normalized Stokes scattering matrix has six independent elements (Paton, 1958; Mishchenko et al., 2002)".

Response: Thanks for your comments. We have revised it in the revised manuscript.

**Comments:** Eq. 3: Provide the assumptions that result in this simplified form of the scattering matrix (randomly-oriented particles etc).

**Response:** Thanks for your comments. We have added the descriptions for the assumptions in the revised manuscript. All the calculations are based on the assumption that dust particles and their mirror counterparts are present in equal numbers in the ensemble of randomly oriented particles. In the atmosphere, it is reasonable to assume that the possibility of each particle direction is identical, which mathematically satisfies the definition of random orientation (Mishchenko and Yurkin, 2017). We have added some descriptions in the revised manuscript:

"All the calculations assume that dust particles are randomly oriented particles (Mishchenko and Yurkin, 2017), and the possibility of each particle direction was assumed to be identical. For the randomly oriented particles, the normalized Stokes scattering matrix has six independent elements:"

**Comments:** Line 130 "...the irregular dust particles.": Substitute with "...the irregular dust particles (Gasteiger et al., 2011)."

**Response:** Thanks for your comments. We have added the reference in the revised manuscript.

**Comments:** Lines 138-139 "...and the accuracy of the DDSCAT is acceptable.": Acceptable based on which threshold? Elaborate and discuss.

**Response:** Thanks for your comments. We have re-written this sentence in the revised manuscript:

"the difference of the scattering matrix of spherical particles calculated using the DDSCAT is below 1%, which is much smaller than the difference caused by the dust shapes. Thus, the accuracy of the DDSCAT is acceptable."

**Comments:** Lines 146-156: Re-do the radiative transfer calculations using the scattering properties of particle ensembles, and not of single particles (see comment (3) above). If the calculations cannot be re-done, omit Sect. 2.4.

**Response:** Thanks for your comments. As responded above, the radiative transfer calculations are indeed unrealistic. However, from theoretical analysis, the radiative transfer calculations for single particles are useful for seeing how the complex shapes of dust with different sizes affect the applicability of spheroids in reproducing the polarized characteristics of irregular dust particles. Thus, we don't delete the radiative transfer calculations. In the future, radiative transfer calculations using realistic size distributions would be conducted.

**Comments:** Line 158 "3.1 Single scattering properties of dust with irregular shapes": Substitute it with "3.1 Single scattering properties of single dust particles with irregular shapes".

**Response:** Thanks for your comments. We have revised it in the revised manuscript.

**Comments:** Line 166 "...would result in more obvious non-sphericity...": What do you mean by this sentence? Please elaborate.

**Response:** Thanks for your comments. We are very sorry for without clarifying clearly the meaning. After careful consideration of your other comments, we have decided to delete the sentence.

**Comments:** Lines 170-179: Different values for R result in quite different shapes. I do not think the comparison presented in this paragraph is very helpful, since you compare almost spheres with spheroids. If you want to keep it, start the paragraph with the last two sentences "When R=0, the binding force..., so  $F_{22}/F_{11}$  become more close to 1."

**Response:** Thanks for your comments. We have revised the manuscript based on your suggestions.

**Comments:** Lines 183-184 "Thus, the spheroid model... for small dust": Substitute with "Thus, single spheroidal particles can provide a reasonable estimation for small single dust particles.". **Response:** Thanks for your comments. We have revised the manuscript based on your suggestions.

**Comments:** Lines 185-187 "Besides, ... fitted using spheroids.": Avoid making these statements. As you can see in Fig. 2 in Gasteiger et al. (2011), more irregular shapes do not necessarily show larger depolarization values (i.e., smaller  $F_{22}/F_{11}$ ).

**Response:** Thanks for pointing it out. We have deleted the statements in the revised manuscript.

**Comments:** Lines 190-191 "The dust... those of spheroids.": This is not generally true either. See again Fig. 2 in Gasteiger et al. (2011).

**Response:** Thanks for pointing it out. We have modified this sentence as "However, the  $F_{22}/F_{11}$  of dust with irregular shapes deviates substantially those of spheroids".

**Comments:** Lines 191-195 "The trends... spheroids.": Please provide quantification instead of "rather similar" and "rather different".

**Response:** Thanks for pointing it out. We have the paragraph:

"With a large particle size, the differences of the scattering matrix of dust with irregular shapes and best-fitted spheroids become rather obvious. The best-fitted spheroids can generally reproduce the F11 trend of dust with irregular shapes, while some obvious differences are observed at certain angles. Figures S4 – S5 show that the absolute  $F_{11}$  relative differences between the irregular dust and the best-fitted spheroids can exceed 75% at backward angles. Besides, the  $F_{22}/F_{11}$  of dust with irregular shapes deviates substantially those of spheroids, and the differences of  $F_{22}/F_{11}$  between the best-fitted spheroids and irregular dust can reach approximately 0.3. The  $F_{33}/F_{11}$  and  $F_{44}/F_{11}$  differences between the best-fitted spheroids and irregular dust are also substantially, which can reach approximately 0.3 and 0.35, respectively (see Figure S5). For the  $F_{12}/F_{11}$  and  $F_{34}/F_{11}$ , the sign can even be modified from negative to oppositive at some scattering angles if substituting the irregular dust with best-fitted spheroids. As shown in Figure S4,  $F_{34}/F_{11}$  and  $F_{34}/F_{11}$  differences of exceeding 0.5 can be observed."

**Comments:** Line 197 "With an original aspect ratio of 1:1, the spheroids…": Substitute with "With an original aspect ratio of 1:1, spheres…".

**Response:** Thanks for your comments. Even with an original aspect ratio of 1:1, the particles can also become non-spherical with the erosion of external force. Thus, we used "spheroids" in the revised manuscript.

Comments: Lines 196-198: Quantify the "relative well" and "relative bad" fits.

**Response:** Thanks for your comments. We have re-written the paragraph:

"Figures 4 – 6 show similar results, but for different original aspect ratios. The original aspect ratio has a signifcant impact on the applicability of spheroids. With an original aspect ratio of 1:1, spheroids ft the scattering matrix of irregular relatively well. As shown in Figure S6, with an original aspect ratio of 1:1, the absolute  $F_{11}$  relative differences between the bestftted spheroids and irregular dust are below 30%, and the differences for other elements are also below 0.3. However, the fits of spheroids are relatively bad for the dust with an original aspect ratio of 2:1 and 1:2 compared to those with an original aspect ratio of 1:1. As shown in Figure S7, the absolute  $F_{11}$  relative differences of approximately 100% between the best-fitted

spheroids and irregular dust are observed. The differences in other elements are also significantly larger than those with original aspect ratio of 1:1. Specifcally, the absolute values of the differences in  $F_{22}/F_{11}$ ,  $F_{33}/F_{11}$  and  $F_{34}/F_{11}$  between the irregular dust particles and best-fitted spheroids could exceed 0.6, 0.8 and 0.6 respectively, when the original aspect ratio is 1:2. The reason may be that the mass of spherical particles is lost relatively uniformly, and the overall structure can be well represented by a spheroid"

Comments: Line 198 "The reason is that...": Substitute with "The reason may be that...".

Response: Thanks for your comments We have revised it in the revised manuscript.

**Comments:** Lines 207-208 "...the best-fitted Cext and Csca...": Do they correspond to the spheroids that best-fit the phase function? Make this more clear.

**Response:** Thanks for your comments. They correspond to the spheroids that best-fit the phase function. We have modified them as "the  $C_{ext}$  and  $C_{sca}$  of best-fitted spheroids" in the revised manuscript.

Comments: Line 210 "...turns obvious...": Substitute with "...increases...".

Response: Thanks for your comments. We have revised it in the revised manuscript.

Comments: Line 212 "...would constraining...": Substitute with "...constrains...".

**Response:** Thanks for your comments. We have revised it in the revised manuscript.

**Comments:** Line 213 "...and the retrieved aspect ratio is more close to 1:1.": Not only the retrieved but also the actual aspect ratio. Correct the sentence accordingly.

Response: Thanks for your comments. We have deleted "retrieved" in the revised manuscript.

Comments: Table 2: Correct the "Aspect Ration" in the header, with "Aspect Ratio".

Response: Thanks for your comments. We have revised it in the revised manuscript.

**Comments:** Section 3.2: Re-calculate the radiative transfer using ensembles of particles, otherwise omit this section (see comment (3) above).

**Response:** Thanks for your comments. As responded in above, the radiative transfer calculations are indeed unrealistic. However, from theoretical analysis, the radiative transfer calculations for single particles are useful for seeing how the complex shapes of dust with different sizes affect the applicability of spheroids in reproducing the polarized characteristics of irregular dust particles. Thus, we don't delete the radiative transfer calculations. In the future, radiative transfer calculations using realistic size distributions would be conducted.

**Comments:** Lines 304-307 "The spheroid model... of complex dust, ...": Substitute with "The spheroidal shapes are commonly used to reproduce the scattering properties of dust, while their applicability is still unclear. To calculate the scattering properties of dust, ....".

**Response:** Thanks for your comments. We have revised it in the revised manuscript.**

**Comments:** Lines 312-314 "... of dust with different shapes.... were compared.": Substitute with "of dust with different shapes (but not for faceted dust particles). To evaluate the capability of spheroids to reproduce the single dust particle scattering properties, we used single spheroidal particles that fit well the phase function of single dust particles with irregular shapes, and then we investigated their capability to reproduce all the elements of the scattering matrix.".

**Response:** Thanks for your comments. We have revised it in the revised manuscript.**

**Comments:** Line 315 "The single... investigated.": Substitute with "The single scattering properties of single dust particles with irregular shapes were investigated.".

**Response:** Thanks for your comments. We have revised it in the revised manuscript.

Comments: Line 318 "...insensitive.": Substitute with "...small." and provide quantification.

**Response:** Thanks for your comments. We have provided quantifications in the revised manuscript:

"With a diameter of 0.2  $\mu$ m, the absolute F11 relative differences between the best-fitted spheroids and irregular dust do not exceed 12%, and the differences of other elements do not exceed 0.05. However, with the particle size increasing, the F11 relative differences of approximately 100% could be observed in certain scattering angles. The maximum differences of other elements between irregular dust particles and best-fitted spheroids can reach approximately 0.3 – 0.8."

Comments: Line 323 "Besides, the sign...": Substitute with "The sign...".

Response: Thanks for your comments. We have revised it in the revised manuscript.

**Comments:** Line 329 ".... close to 1:1.": Substitute with "... close to 1:1, and the particles become more spherical.".

Response: Thanks for your comments. We have revised it in the revised manuscript.

**Comments:** Lines 330-340: Re-calculate the radiative transfer using ensembles of particles, otherwise omit this part (see comment (3) above).

**Response:** Thanks for your comments. As responded in above, the radiative transfer calculations are indeed unrealistic. However, from theoretical analysis, the radiative transfer calculations for single particles are useful for seeing how the complex shapes of dust with different sizes affect the applicability of spheroids in reproducing the polarized characteristics of irregular dust particles. Thus, we don't delete the radiative transfer calculations. In the future, radiative transfer calculations using ensembles of particles with realistic size distributions would be conducted.

**Supporting Information for "The polarimetric characteristics of dust with irregular shapes: Evaluation of the spheroid model"**

Jie Luo1, Zhengqiang Li1,2, Cheng Fan1, Hua Xu1,2, Ying Zhang1,2, Weizhen Hou1,2, Lili Qie1, Haoran Gu1,2,3, Mengyao Zhu1,2,4, Yinna Li1,2, and Kaitao Li1

1State Environment Protection Key Laboratory of Satellite Remote Sensing, Aerospace Information Research Institute, Chinese Academy of Sciences, Beijing 100101, China

2University of Chinese Academy of Sciences, Beijing 100049, China

3College of Geography and Tourism, Anhui Normal University, Wuhu 241003, China

4College of Geoexploration Science and Technology, Jilin University, Changchun 130026, China

Correspondence: Zhengqiang Li (lizq@radi.ac.cn)

Figure S1. The scattering matrix of spherical particles calculated using the DDSCAT and T-matrix codes, respectively, where  $d_p = 0.4 \ \mu \text{m}$ .